# Effectiveness of a Judo Intervention Programme on the Psychosocial Area in Secondary School Education Students

**DOI:** 10.3390/sports11080140

**Published:** 2023-07-28

**Authors:** David Lindell-Postigo, Félix Zurita-Ortega, Eduardo Melguizo-Ibáñez, Gabriel González-Valero, Manuel Ortiz-Franco, José Luis Ubago-Jiménez

**Affiliations:** 1Department of Physical Education, Sunland International School, 29580 Málaga, Spain; 2Department of Didactics of Musical, Plastic and Corporal Expression, University of Granada, 18071 Granada, Spainggvalero@ugr.es (G.G.-V.); jlubago@ugr.es (J.L.U.-J.); 3La Inmaculada Teacher Training Centre, 18013 Granada, Spain

**Keywords:** Judo, sport motivation, self-concept, emotional intelligence, aggressive behaviour

## Abstract

Currently, many combat sports are pedagogically conceived as uneducational and unreliable for the development of young people. The present research aims to investigate the influence of a Judo intervention programme on the motivational climate towards sport, aggressive behaviour, emotional intelligence, and self-concept in secondary school students and to establish the relationships between them. This objective was broken down into (a) developing an explanatory model of the variables mentioned above and (b) testing the model equations through a multi-group analysis in terms of pre-test and post-test. The present study conducted a pre–post-test quasi-experimental design with a single experimental group. The sample consisted of a total of 139 adolescents (12.67 ± 1.066), 50.4% of whom were male (*n* = 70) and 49.6% female (*n* = 69). The results show that the intervention decreased all types of aggression and increased levels of emotional intelligence. An increase in social, physical and academic self-concept and decreases in the family and emotional areas were also observed. Finally, for the motivational climate, a tendency towards the ego climate to the detriment of the task climate was observed. It is concluded that the Judo intervention programme is effective in decreasing aggressive behaviour and effective in increasing levels of emotional intelligence and self-concept.

## 1. Introduction

Violence is said to be a construct deeply included in people’s life, which leads the causes of death worldwide for people aged 15–44 years [1]; it is associated with disruptive and harmful behaviours [2], and has a negative impact on people’s welfare [3,4]. Also, it is correlated to psychosocial factors, as Turner et al. [5] concluded in the longitudinal study they conducted, in which they found a negative association between poly-victimisation and youth mental health. Thus, despite its uncertain origin [6,7], it is essential to address it to enhance society’s welfare [2,8], also considering the possible savings in health systems due to the impact of violence during childhood on adulthood [2,9].

Following on from the foregoing, violence seems to improve the aggressor’s physical self-concept, whereas it is detrimental for all the dimensions of a victim’s self-concept [4]. Nevertheless, improving individual’s self-concept seems to buffer aggression, since it reduces both violence [5,10,11] and victimisation [12]. Lastly, Blakely-McClure and Ostrov [13] concluded that relational aggression reduces academic self-concept but has a positive impact on the physical one, albeit higher levels of self-concept may also lead to decreased relational aggression.

Plus, motivational climate also influences disruptive behaviours, since a task-oriented climate is positively correlated to prosocial behaviours, reducing violence, whereas an ego-oriented (EC) is positively associated with antisocial behaviours, increasing violence [14,15,16,17], just like victimisation [18]; Castro-Sánchez et al. [18] pointed out that this association is stronger in sedentary children. Miller et al. [19] explained this association as being due to predictions based on a lower capacity for moral judgment within the goal-oriented climate.

Regarding the relationship between violence and emotional intelligence (EI), the systematic review conducted by Estévez et al. [4] confirmed the negative association between both constructs regardless of the type of violence, which was explained by them and Kokkinos and Kipritsi [20] as being because children able to manage emotions are prone to being more socially accepted by their peers. Moreover, Kokkinos and Kipritsi [20] found that EI becomes a predictor of victimisation. Further, Baroncelli and Ciucci [21] found that low levels of EI are linked to cybebullying, especially in terms of the regulation of emotions. In line with this, Castillo et al. [22] successfully implemented a Socio-Emotional Learning programme to reduce aggressive behaviours in adolescents, especially with regard to the emotional aspects of aggression, which highlights the buffering role played by this construct in reductions in violence.

There is a scarcity of studies concerning how all these constructs are associated with each other, although some articles shed light on the association among self-concept, EI and violence. For instance, Cañas et al. [23] found a negative association of cyberbullying with SC—especially family and academic dimensions—and with emotional intelligence—especially emotional regulation and emotional comprehension. Further, Cañas et al. [3] found the moderating effect that EI has in the harmful relationship between victimisation and SC.

Another factor to consider in this study is the practice of physical activity and its relationship with these psychosocial factors, specifically the role of Physical Education and martial arts. As a matter of fact, some pieces of research have revealed the positive effects of this practice on disruptive behaviours [24,25] or task climate [26], the latter enabling individuals to develop more adaptive motivational patterns [26] and engaging people in sport more favorably due to enjoyment [27]. On the other hand, despite it being unclear whether physical activity improves individuals’ self-concept or whether people with higher levels of self-concept engage more often in physical activity [28], physical self-concept seems to be crucial in people’s engagement in physical exercise [28,29,30], even in harmful situations such as a lockdowns [29,31]; this was confirmed by Conant et al. [32] and Rivera et al. [33], who conducted two different studies using martial arts with people with special needs, with positive effects shown on their self-concept. Regarding violence, Zillmann et al. [34] claimed that the intensity of physical exercise is positively correlated to aggressive behaviours, while Viner et al. [35] confirmed the mediating effect of physical exercise on cyberbullying and mental health.

Additionally, Judo is conceived as a martial art and combat sport, with a philosophy initially built around violence [36], and it is a physical activity attracting increasing interest among people [37]. These disciplines are said to be violent activities undertaken by aggressive people [38]. However, this violence in modern martial arts is powered by competitiveness [39]. Indeed, experts usually recommend the practice of martial arts due to its benefits on people’s health [40,41], even as a form of psychotherapy [2,42,43].

Regarding the association between the practice of martial arts and these constructs, martial artists tend to have higher levels of self-concept, especially physically [31]. Further, Judokas tend to have higher emotional intelligence in all its dimensions except emotional attention [44], although no significant differences exist between Judo and other martial arts [45]. Additionally, Zurita-Ortega et al. [46] found a tendency to shift the motivational climate towards EC as the level of competition increases, which leads to an increase in aggression, as Iwasaki et al. [47] reported. Overall, a task climate addressed through Judo has been effective in reducing bullying and attitudes towards violence [48].

Consequently, conducting an intervention programme is essential to estimating the effectiveness of the theoretical framework, especially in terms of educational programmes, in which experts need to sift through several hypothetical methodologies. Hence, researchers have produced many articles in this field over the years, aimed at shedding light on how school-based intervention programmes positively affect individuals’ welfare [49], especially in primary schools, since this is the stage at which violent behaviours begin [50]. In this line, it is explained that a proactive intervention is much more useful to reducing bullying, especially when it is focused on the role of bystanders and peer tutoring [51]. Plus, some research found a positive impact of intervention programmes at school on individuals’ welfare by reducing violence [52].

Little research exists regarding how conducting a Judo intervention programme may influence these constructs; on this matter, Montero-Carretero et al. [48] successfully undertook one throughout ten PE sessions, twice a week, to reduce bullying. Nevertheless, widening the field of research, a martial art intervention has been proven to be effective to this end, especially in terms of reducing disruptive behaviours [41].

Therefore, this study aims to further investigate the influence of a Judo intervention programme on self-concept, motivational climate, emotional intelligence and aggressive behaviours in secondary school students in a Spanish school, and to establish the relationships between them. The last goal breaks down the task into (a) developing an explanatory model of the variables mentioned above and (b) testing the model equations through multigroup analysis in terms of pre-test and post-test.

## 2. Materials and Methods

### 2.1. Participants

The present study exhibits a pre–post-test quasi-experimental design with a single experimental group. The sample comprised 70 (50.4%) male and 69 (49.6%) female adolescents aged 11–16 (12.67 ± 1.06), who attended a secondary school in Andalusia, specifically in Malaga city (Spain). Voluntary response sampling to select participants was conducted, ensuring that they had no unsolved questions and understood how to code themselves. Lastly, the inclusion criteria were for the participants to be in full control of their mental faculties and to possess informed consent given by their parent/guardian.

### 2.2. Instruments and Variables

This section describes the instruments used to collect the variables.

**An ad-hoc questionnaire** was used to collect socio-demographic characteristics, which included gender (male/female), age, whether subjects regularly practised physical activity (that is, more than three hours a week according to World Health Organization recommendations [53]), whether that practice involved martial arts, whether they competed and level of competition (local, regional, national, or international).

**Self-concept Form-5** was used to evaluate the self-concept variable. Based on Shavelson et al. [54], this instrument was created by Garcia and Musitu [55], yet we used an adaptation for Judoists developed by Zurita-Ortega et al. [56]. Thus, it comprises 24 items grouped into five dimensions: academic self-concept (AC: items 1, 6, 11, 16, 21, 26), social self-concept (SO: items 2, 7, 12, 17, 22, 27), emotional self-concept (EM: items 3, 8, 13, 18, 23, 28), familiar self-concept (FA: items 4, 9, 14, 19, 24, 29) and physical self-concept (PH: items 5, 10, 15, 20, 25, 30). It measures SC using a five-level Likert scale (1 = never and 5 = always). For the pre-test a reliability level of α = 0.731 was obtained, while for the post-test a value of α = 0.769 was achieved.

**Perceived Motivational Climate in Sport Questionnaire-2 (PMCSQ-2)** was used to evaluate motivational climate among students. This questionnaire was designed by Newton et al. [57], yet we used the adapted version proposed by Gonzalez-Cutre et al. [58], made up of 33 items rated on a five-level Likert scale (1 = totally disagree and 5 = totally agree), which evaluates motivation within two dimensions: TC (which comprises three sub-dimensions: effort, improvement and cooperative learning), and EC (which is composed of three sub-dimensions: unequal recognition, punishment for mistakes and intra-team rivalry). During the pre-test, the Cronbach’s alpha obtained a value of 0.954 for the task climate and 0.962 for the ego climate. In the post-test, Cronbach’s alpha obtained a value of 0.939 for the task climate and a value of 0.949 for the ego climate.

**Schutte Self-Report Inventory (SSRI)** was used to evaluate emotional intelligence. Specifically, we employed the Spanish adaptation from García-Coll et al. [59] of the test created by Schutte et al. [60]. The adaptation removed three items due to their negative formulation, finally resulting in 30 items. Hence, the questionnaire measures five dimensions of emotional intelligence using a five-level Likert scale (1 = totally disagree and 5 = totally agree), specifically focusing on emotion perception (EP: items 2, 3, 9, 11, 13, 20 and 29), self-emotional management (SEM: items 8, 14, 17, 18, 21, 24, 27 and 30), heteroemotional management (HEM: items 1, 4, 5, 7, 10, 12, 15, 23, 25 and 28), emotion utilisation (EU: items 6, 16, 19 and 26) and general emotional intelligence (sum of all items). In the pre-test, the following internal reliabilities were estimated: emotion perception α = 0.885, self-emotional management α = 0.863, self-emotional management α = 0.851, emotion utilisation α = 0.873 and general emotional intelligence α = 0.889. In the post-test the following internal reliabilities were estimated: emotion perception α = 0.834, self-emotional management α = 0.869, self-emotional management α = 0.849, emotion utilisation α = 0.837 and general emotional intelligence α = 0.882.

**The School Violence Scale** was employed to evaluate individuals’ aggressive behaviours. We used the Spanish version employed by Musitu et al. [61] from the School Violence Scale (SVS), which was designed by Little et al. [62]. This questionnaire evaluates two types of violence using a four-level Likert scale (1 = Never and 4 = Always), namely, over/direct aggression (OA) and relational/indirect aggression (RA); for this aim, the questionnaire comprises 25 items. In the pre-test, the following internal reliabilities were estimated: over/direct aggression α = 0.929, relational/indirect aggression α = 0.919 and general violence α = 0.921. In the post-test, the following internal reliabilities were estimated: over/direct aggression α = 0.920, relational/indirect aggression α = 0.925 and general violence α = 0.926

### 2.3. Procedure

Prior to running the intervention programme, which fulfils the requirements of the Research Ethics Committee of the University of Granada, coded 2966/CEIH/2022, and the Declaration of Helskinki, the researchers contacted the Principal and Physical Education teachers to fully explain both its aims and purposes. After obtaining their authorisation to perform the research at school, students were contacted through their physical education teachers. Anonymously, the voluntariness of participation was given, and data treatment only for scientific purposes was assured. Then, a booklet with all the instruments, which was created by the Department of Didactics of Musical, Plastic and Corporal expression of the University of Granada, was distributed before and after the intervention programme. Moreover, two-thirds of the teachers were trained to teach a Judo topic by one of the researchers, who is a Judo Master, due to their lack of knowledge.

The training received by teachers focused on the techniques and dynamics of Judo classes. The training period for the teachers lasted three months. Regarding the technical training, the teachers were trained in the analysis and performance of different keys (O-Soto-Gari, Yoko-Shiho-Gatame, Tate-Shiho-Gatame and Uki-Goshi). Regarding class dynamics, the teachers were trained in Judo didactics, receiving classes from a second Dan belt, specialised in the didactics of this sport. The teachers also received three hours of Judo training per week. The first month was focused on knowledge related to ground immobilisations and ground Judo sparring. The second month was focused on the teaching of falling and throwing techniques. During the second month, one hour was dedicated per week to reminding the participants of the knowledge of ground Judo. Finally, the third and last month focused on Judo foot and Judo ground sparring. It is also worth mentioning that during the last month, one hour a week was dedicated to creating checklists together with the Judo teaching expert to evaluate the students during the intervention programme. The teachers were trained in a gymnasium in the province of Granada, which had its own tatami. Likewise, the intervention programme was based on the curriculum that governs compulsory secondary education in Spain, so that this intervention programme was contextualised within Spanish educational law. The implementation of the topic took eight lessons, lasting two months, and was supervised by researchers to ensure that it was carried out correctly. Data were collected before and after the intervention, during one of the students’ PE lessons.

The intervention programme was based on regular Judo lessons, beginning with a warm-up consisting of pulse-raiser games to improve mobility and displacements (5 min), followed by technique drills/games (in which individuals practiced fall techniques, holding techniques, gripping, and displacing the opponent and throwing techniques), and finishing with matches (40 min), questions and answers and stretching (5 min). During the week, two sessions of 50 min each were held. All sessions were held on the Tatami of the school. Previously, it was observed that the Tatami was in good condition and guaranteed a reliable degree of safety. Due to the type of sport that Judo is, three legal guardians did not authorize their children to participate in this intervention programme. The inclusion criteria established consisted of being between 11 and 16 years of age and being authorised by the legal guardians to participate in this intervention program.

The time distribution of the intervention programme was ensured as follows:**Session 1 to Session 4:** Games related to immobilising the opponent on the ground were carried out.**Session 4 to Session 10:** Ground Judo techniques were taught. Also, during session 9 and session 10 the students practiced randori on the ground using some immobilisation.**Session 11:** A day of technical evaluation of the immobilisations. The assessment consisted of performing two ground immobilisations on a colleague.**Session 12 to Session 16:** Playing games to teach the different falls.**Session 16 to Session 20:** Teaching Judo foot techniques.**Session 20 to Session 23:** Randori on foot applying the techniques seen from session 16 to session 20.**Session 24:** Evaluation of the technical execution of falls and Judo foot. The assessment consisted of performing a drop and a Judo foot technique.

This intervention programme had the following objectives:To learn the techniques of the main ground Judo and foot Judo keys;To apply Judo ground and Judo foot techniques to randori situations.

However, due to national restrictions related to the COVID pandemic, all students had to always wear masks during the intervention to ensure their welfare. Finally, this intervention programme was carried out in a school in the province of Malaga, during the months of January and March 2022.

### 2.4. Data Analysis

Data were processed with IBM SPSS Statics 25.0 (IBM Corp., Armonk, NY, USA). A correlational study was carried out by performing the Student T-test to find statistically significant differences between pre-intervention and post-intervention results; statistical differences were estimated through Pearson’s Chi-square test (*p* ≤ 0.05). The magnitude of difference in effect size (ES) was obtained with Cohen’s standardised d-index [63], interpreted as null (0.0–0.19), small (0.20–0.49), medium (0.50–0.79), and large (≥0.80). Normality and homogeneity variance were estimated by performing the Kolmogorov–Smirnov test, in which a normal distribution was found.

Structural equation modeling (SEM) was undertaken via IBM SPSS Amos 26.0 (IBM Corp., Armonk, NY, USA) to establish the association among the variables that compound the theoretical model (Figure 1). In this study, a structural equation model was created regarding pre-test and post-test situation. Each one is made up of two exogenous variables (TC; EC) and eighteen endogenous (AC; SO; FA; EM; PH; ROA; POA; IOA; IRA; RRA; PRA; EU; HM; SEM; EP; SC; VI; EI). The latter group was causally explained considering the observed associations between index and measurement reliability, so a measurement error was included. Similarly, the unidirectional arrows represent the influence lines between latent variables and are interpreted from regression weight. Pearson’s Chi-squared test established a significance level of 0.05.

Lastly, the model was evaluated by estimating the component parameters. According to the established criteria [64,65], the Chi-squared goodness-of-fit test was performed and indicated the good adjustment of the model. The comparative fit index (CFI) has a value over 0.95, which indicates a good adjustment to the model. The goodness-of-fit index (GFI) took a value over 0.900, which indicates an acceptable adjustment. The incremental fit index (IFI) took a value over 0.90, which indicates an acceptable adjustment. Root mean square error of approximation (RMSEA) took a value below 0.100, which indicates an acceptable adjustment to the model.

## 3. Results

Table 1 displays the descriptive analysis of the sample, which was made up of 50.4% (*n* = 70) male and 49.6% (*n* = 69) female students. Whilst 77% of the sample regularly practised physical activity according to WHO recommendations [53], 23% did not. Further, 14.4% of individuals practised martial arts before the implementation of the intervention programme, whereas 85.6% of them had never practised any type of martial arts.

Table 2 displays the effects of the intervention programme, demonstrating statistically significant differences (*p* ≤ 0.05). Regarding self-concept, there is a detrimental effect in family self-concept (*p* ≤ 0.05) and emotional self-concept (*p* ≤ 0.05), although there is an improvement in social self-concept, physical self-concept (*p* ≤ 0.05) and academic self-concept. Regarding motivational climate in sports, task climate was reduced after the intervention and ego climate levels increased. Furter, there was a decrease in reactive over aggression, pure over aggressiveness, instrumental relational aggression, pure relational aggression (*p* ≤ 0.05), reactive relational aggression and instrumental relational aggression. Regarding emotional intelligence, there was an increase in emotional perception, self-emotional management, heteroemotional management and emotional utilisation.

Table 3 displays the regression weights of the proposed theoretical model (pre-test), with statistically significant associations shown as *p* ≤ 0.05 and *p* ≤ 0.001. Self-concept is positively associated with academic dimension (β = 0.612), social area (*p* ≤ 0.001; β = 0.507), family dimension (*p* ≤ 0.001; β = 0.520), physical dimension (*p* ≤ 0.001; β = 0.668) and task climate (*p* ≤ 0.05; β = 0.880); nevertheless, it is negatively correlated with ego climate (β = −0.580), violence (*p* ≤ 0.05; β = −0.384) and emotional intelligence (*p* ≤ 0.001; β = −0.487). Further, emotional intelligence is positively associated with ego climate (β = 0.283), emotional utilisation (β = 0.186), heteroemotional management (*p* ≤ 0.05; β = 0.674) self-emotional management (*p* ≤ 0.05; β = 0.791) and emotional perception (*p* ≤ 0.05; β = 0.863), although it is negatively correlated with task climate (β = −0.038) and violence (β = −0.298). Regarding this last variable, it is positively associated with ego climate (β = 0.080), reactive over aggressiveness (*p* ≤ 0.001; β = 0.711), instrumental over aggressiveness (*p* ≤ 0.001; β = 0.855), pure relational aggression (*p* ≤ 0.001; β = 0.569), reactive relational aggression (*p* ≤ 0.001; β = 0.846), instrumental relational aggression (β = 0.729), pure relational aggression (*p* ≤ 0.001; β = 0.855) and task climate (β = −0.509). Lastly, there is a negative association between both motivational climates (*p* ≤ 0.05; β = −0.286).

To focus on the designed model as regards the post-test, the Chi-squared analysis indicates a significant value (X^2^ = 39.764; df = 20; pl = 0.007). CFI had a value of 0.939, NFI scored 0.996, IFI scored 0.943, and the Tucker–Lewis index (TLI) was 0.934. Lastly, the RMSEA scored 0.049.

Table 4 shows the regression weights of the proposed theoretical model (post-test), with statistically significant associations shown as *p* ≤ 0.05 and *p* ≤ 0.001. Self-concept was positively associated with academic area (β = 0.606), social dimension (*p* ≤ 0.001; β = 0.512), family area (*p* ≤ 0.001; β = 0.581), physical dimension (*p* ≤ 0.001; β = 0.658) and task climate (*p* ≤ 0.001; β = 0.549); nevertheless, it was negatively correlated with emotional self-concept (*p* ≤ 0.001; β = −0.438), violence (β = −0.130) and ego climate (*p* ≤ 0.05; β = −0.340). Emotional intelligence was positively associated with ego climate (β = 0.018), emotional utilisation (β = 0.166), heteroemotional management (β = 0.694), self-emotional management (β = 0.849), emotional perception (β = 0.801) and task climate (β = 0.005). On the other hand, it has a negative association with violence (*p* ≤ 0.05; β = −0.304). Further, violence is positively correlated with pure over aggression (*p* ≤ 0.001; β = 0.704), instrumental over aggression (*p* ≤ 0.001; β = 0.851), pure relational aggression (*p* ≤ 0.001; β = 0.562), reactive relational aggression (*p* ≤ 0.001; β = 0.841), instrumental relational aggression (β = 0.723), pure relational aggression (*p* ≤ 0.001; β = 0.629) and ego climate (β = 0.098). In opposition, violence is negatively related to task climate (β = 0.013). Lastly, a negative association was found between both motivational climates (*p* ≤ 0.05; α = −0.336).

## 4. Discussion

This research shows the results of an intervention programme based on Judo for the improvement of psychosocial aspects. The reasons for the application of this sport lie in the mental benefits and increased motivation towards sport [48].

The eminently descriptive data show that the adolescent population analysed was more active than other populations analysed (66.2% practice more than 3 h of physical activity per week) [31]. It is observed that some of the students surveyed practice martial arts, which highlights the increased levels of contact sports in the adolescent population (14.5% practice martial arts) [31]. Even though a large part of the population conceived contact sports as highly injurious, the adequate teaching of falls in the initiation stage helps to prevent injuries [31,66,67].

Continuing with the comparative analysis of the intervention programme, it is observed that the Judo intervention is beneficial for increasing the social, physical and academic areas of self-concept. These results suggest that the practice of Judo helps to improve some dimensions of self-concept. It has been observed that the regular practice of sport helps to improve social recognition [67]. It has also been observed that the regular practice of physical exercise helps to improve the physical image of the person, and these results coincide with others already received [37]. Also, the academic area has been improved, affirming that the regular practice of physical exercise helps to improve the academic perspective [31,33].

On the contrary, a decrease in the levels of the family and emotional area has been observed. These results are quite novel, since the family area promotes the practice of physical activity during the adolescent stage [31]; however, there may be a prior notion that Judo is a dangerous sport that is harmful [66,67]. Continuing with the effects of the motivational climate, a decrease in the task climate and an increase in the levels of the ego climate were observed. These results are due to the sport modality practiced [37], since, as an individual sport that requires contact to defeat the opponent, it requires a high degree of technique and competence [11,31,50].

The decrease in the levels of the emotional area suggests results very distant from those of other investigations, since Judo is characterised as a sport that requires a high level of emotional management [37,68]. The intervention programme has been effective in improving all dimensions of emotional intelligence. Given these findings, it has been observed that other students show an improvement in emotional intelligence through contact sports [68].

Through contact sports, the improvement in emotional intelligence is due to the importance of emotional intelligence during the practice of this type of sport [69]. The comparative analysis suggests a decrease in the levels of violence in all its dimensions. In view of these findings, Antoñanzas [69], Bibi et al. [70] and Estévez-Casellas et al. [4] explained the buffering effect of emotional intelligence on aggressive behaviours in adolescents; in addition, Alvarado et al. [71] and Segura et al. [72] stated that children with higher emotional intelligence tend to show a reduced probability of aggressive behaviours.

Continuing with the analysis based on structural equation modeling, the intervention programme reduced the effect of student violence on emotional intelligence and task climate. This finding is in line with many studies that have studied the reducing effect of the intervention on disruptive behaviours [52,67,73], and it effect of increasing task climate [66] and emotional intelligence [74]. Focusing on martial arts-based intervention programmes, the results are in line with those of Montero-Carretero et al. [48] and Twemlow et al. [7], who also found a reductive effect on aggressive behaviours after their interventions.

Focusing on the practical and legal feasibility of this intervention programme, the current curriculum of the secondary education stage suggests that sports initiation should be carried out through games [75]. Likewise, it is important to achieve the development not only of the motor competence of young people, but also critical analysis is necessary to strengthen attitudes and values related to the body, movement, and the relationship with the environment. The characteristics of a game centred on sports involve the integral development of the person, since different areas of the psychosocial profiles of young people are promoted [76,77,78].

Although the research has fulfilled the proposed objectives, it is not free of limitations. The first limitation is related to the sample, since the intervention programme was developed in only one educational center. Also, although the teachers were trained in Judo teaching and evaluation, they were not experts. It should be noted that a control group and an experimental group were not used, but only one intervention group. Despite this, the results are completely reliable, since fully validated instruments adapted to the sample were used. Regarding future perspectives, this study allows us to understand that Judo is not a sport that encourages violent behaviour, but a sport that allows us to improve the psychosocial area. It would be interesting to repeat this intervention programme including a control group and an experimental group. It would be convenient to carry out the intervention in other schools. This research shows that Judo has a place in the secondary school curriculum. Although this sport can be taught in public schools, it is advisable to check whether the equipment in the school offers safety during the learning of the sport. If this element is fulfilled, it would be interesting to develop a one-term learning situation through which elements related to ground Judo and standing Judo can be taught.

Finally, regarding the practical applications of this research, it can be highlighted that from these results a learning situation can be designed that includes the primary and secondary education stages. In the primary education stage, Judo contents would be approached through games involving grappling and ground Judo. Subsequently, in the secondary school stage, we will begin to practice Judo foot through games and the approach of basic and elementary keys. For future intervention programmes, it would be interesting to add breakfalls.

## 5. Conclusions

This intervention programme has shown that Judo is a sport that can be used to improve the psychosocial environment of Spanish adolescent high school students.

It has been observed that a Judo programme contextualised within the current educational regulations is useful to improve self-concept, specifically the physical dimension of self-concept. Variations have been observed in the motivational climate towards which the sport practice is oriented, since after the intervention the effects of the task climate increased positively, decreasing the effects of the ego climate. Moreover, improvements were also obtained in the emotional area of the participants. In addition, the levels of aggressiveness of the students decreased.

As a general result, it has been observed that Judo is a very useful sport for achieving the integral education of the students and reducing aggressive behaviour in this population.

## Figures and Tables

**Figure 1 sports-11-00140-f001:**
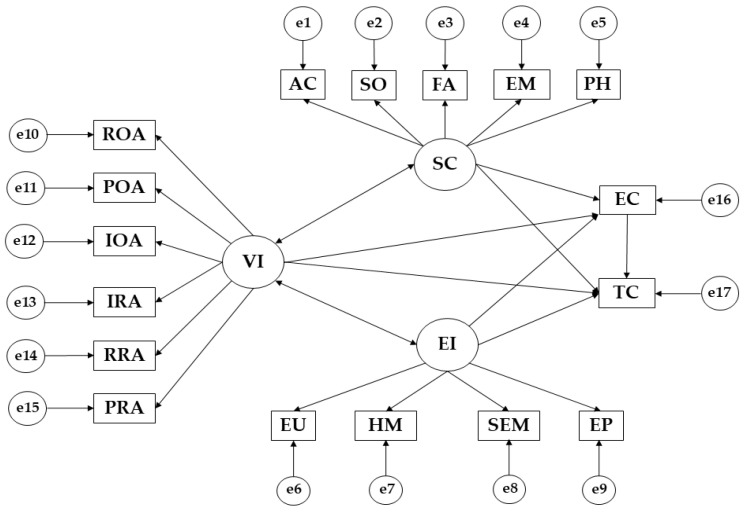
Proposed structural equational modeling. Note: Family Self-Concept (FA); Social Self-Concept (SO); Physical Self-Concept (PH); Emotional Self-Concept (EM); Academic Self-Concept (AC); Task Climate (TC); Ego Climate (EC); Reactive Over Aggression (ROA); Pure Over Aggression (POA); Instrumental Over Aggression (IOA); Pure Relational Aggression (PRA); Reactive Relational Aggression (RRA); Instrumental Relational Aggression (IRA); Emotional Perception (EP); Self- Emotional Management (SEM); Heteroemotional Management (HM); Emotional Utilisation (EU); Emotional Intelligence (EI); Violence (VI); Self-Concept (SC).

**Table 1 sports-11-00140-t001:** Descriptive data of the sample.

		*n*	%
**Gender**	**Male**	70	50.4%
	**Female**	69	49.6%
**Do you practice more than 3 h of PA per week outside school hours?**	**Yes**	107	77.0%
	**No**	32	23.0%
**Do you practice any kind of martial arts?**	**Yes**	20	14.4%
	**No**	119	85.6%

**Table 2 sports-11-00140-t002:** Comparative pre-test post-test study.

Variable	Test	Mean ± Standard Deviation	T	*p*	*d*	95% Confidence Interval
Family Self-Concept	Pre	3.61 ± 0.46	0.224	≤0.05	0.022	[0.213; 0.257]
	Post	3.60 ± 0.46				
Social Self-Concept	Pre	3.53 ± 0.46	−1.481	˃0.05	0.137	[0.098; 0.372]
	Post	3.59 ± 0.40				
Physical Self-Concept	Pre	3.48 ± 0.86	−3.248	≤0.05	0.181	[0.055; 0.416]
	Post	3.63 ± 0.79				
Emotional Self-Concept	Pre	2.77 ± 0.87	2.322	≤0.05	0.125	[0.111; 0.36]
	Post	2.66 ± 0.89				
Academic Self-Concept	Pre	3.96 ± 0.60	−0.566	˃0.05	0.032	[0.203; 0.267]
	Post	3.98 ± 0.64				
Task Climate	Pre	4.08 ± 0.51	1.561	˃0.05	0.123	[0.113; 0.358]
	Post	4.01 ± 0.62				
Ego Climate	Pre	2.30 ± 0.61	−1.933	˃0.05	0.149	[0.086; 0.385]
	Post	2.40 ± 0.71				
Reactive Over Aggression	Pre	1.09 ± 0.52	0.707	˃0.05	0.059	[0.177; 0.294]
	Post	1.08 ± 0.50				
Pure Over Aggression	Pre	1.52 ± 0.35	0.406	˃0.05	0.033	[0.202; 0.269]
	Post	1.49 ± 0.22				
Instrumental Over Agression	Pre	1.39 ± 0.42	1.919	˃0.05	0.149	[0.086; 0.385]
	Post	1.33 ± 0.37				
Pure Relational Aggression	Pre	1.60 ± 0.49	3.112	≤0.05	0.258	[0.022; 0.494]
	Post	1.48 ± 0.43				
Reactive Relational Aggression	Pre	1.17 ± 0.34	0.503	˃0.05	0.065	[0.17; 0.3]
	Post	1.15 ± 0.26				
Instrumental Relational Aggression	Pre	1.29 ± 0.39	0.301	˃0.05	0.026	[0.209; 0.261]
	Post	1.28 ± 0.38				
Emotional Perception	Pre	2.82 ± 0.51	−1.007	˃0.05	0.077	[0.158; 0.312]
	Post	2.86 ± 0.52				
Self-Emotional Management	Pre	2.59 ± 0.38	−0.647	˃0.05	0.048	[0.187; 0.284]
	Post	2.61 ± 0.43				
Heteroemotional Management	Pre	2.95 ± 0.37	−0.276	˃0.05	0.027	[0.208; 0.263]
	Post	2.96 ± 0.35				
Emotional Utilisation	Pre	2.30 ± 0.45	−0.052	˃0.05	0.004	[0.231; 0.239]
	Post	2.34 ± 0.43				

**Table 3 sports-11-00140-t003:** Regression weights before applying the intervention programme.

Direction	Regression Weights				Standardised Regression Weight
	Estimation	Estimation Error	Critical Ratio	*p*	Stimations
EC ← SC	−1.055	0.558	−1.891	0.059	−0.580
EC ← EI	2.500	2.697	0.927	0.354	0.283
EC ← VI	0.206	0.266	0.775	0.438	0.080
AC ← SC	1.000				0.612
SO ← SC	0.519	0.103	5.031	***	0.507
FA ← SC	−1.100	0.226	−4.869	***	−0.487
EM ← SC	0.608	0.118	5.141	***	0.520
PH ← SC	1.351	0.216	6.249	***	0.668
EU ← EI	1.000				0.186
HM ← EI	2.902	1.432	2.026	**	0.674
SEM ← EI	4.251	2.077	2.047	**	0.791
EP ← EI	5.589	2.722	2.053	**	0.863
IRA ← VI	1.000				0.729
RRA ← VI	0.798	0.084	9.526	***	0.846
PRA ← VI	0.993	0.139	7.146	***	0.636
IOA ← VI	1.154	0.120	9.618	***	0.855
ROA ← VI	0.566	0.071	8.021	***	0.711
POA ← VI	1.034	0.162	6.390	***	0.569
TC ← SC	1.584	0.576	2.748	**	0.880
TC ← EI	−3.892	2.958	−1.316	0.188	−0.509
TC ← VI	0.084	0.216	0.389	0.697	0.038
TC ← EC	−0.248	0.086	−2.881	**	−0.286
VI ←→ EI	−0.007	0.004	−1.682	0.093	−0.298
VI ←→ SC	−0.042	0.013	−3.126	**	−0.384

Note: Family Self-Concept (FA); Social Self-Concept (SO); Physical Self-Concept (PH); Emotional Self-Concept (EM); Academic Self-Concept (AC); Task Climate (TC); Ego Climate (EC); Reactive Over Aggression (ROA); Pure Over Aggression (POA); Instrumental Over Aggression (IOA); Pure Relational Aggression (PRA); Reactive Relational Aggression (RRA); Instrumental Relational Aggression (IRA); Emotional Perception (EP); Self-Emotional Management (SEM); Heteroemotional Management (HM); Emotional Utilisation (EU); Emotional Intelligence (EI); Violence (VI); Self-Concept (SC). Note: ** *p* ≤ 0.05; *** *p* ≤ 0.001.

**Table 4 sports-11-00140-t004:** Regression weights after applying the intervention programme.

Direction	Regression Weights				Standardised Regression Weight
	Estimation	Estimation Error	Critical Ratio	Estimation	Estimation Error
EC ← SC	−0.626	0.205	−3.046	**	−0.340
EC ← EI	0.259	0.246	1.050	0.294	0.018
EC ← VI	0.181	0.887	0.204	0.839	0.098
AC ← SC	1.000				0.606
SO ← SC	0.530	0.114	4.643	***	0.512
FA ← SC	0.685	0.135	5.086	***	0.581
EM ← SC	−0.997	0.243	−4.099	***	−0.438
PH ← SC	1.342	0.244	5.505	***	0.658
EU ← EI	1.000				0.166
HM ← EI	3.362	1.889	1.780	0.75	0.694
SEM ← EI	5.135	2.866	1.792	0.73	0.849
EP ← EI	5.842	3.263	1.790	0.073	0.801
IRA ← VI	1.000				0.723
RRA ← VI	0.797	0.086	9.309	***	0.841
PRA ← VI	0.994	0.142	6.998	***	0.629
IOA ← VI	1.155	0.123	9.404	***	0.851
ROA ← VI	0.566	0.072	7.844	***	0.704
POA ← VI	1.035	0.165	6.258	***	0.562
TC ← SC	0.873	0.181	4.819	***	0.549
TC ← EI	0.040	0.596	0.067	0.947	0.005
TC ← VI	0.030	0.171	0.176	0.860	0.013
TC ← EC	−0.290	0.063	−4.621	***	−0.336
VI ←→ EI	−0.032	0.013	−2.582	**	−0.304
VI←→ SC	−0.003	0.002	−1.081	0.280	−0.130

Note: Family Self-Concept (FA); Social Self-Concept (SO); Physical Self-Concept (PH); Emotional Self-Concept (EM); Academic Self-Concept (AC); Task Climate (TC); Ego Climate (EC); Reactive Over Aggression (ROA); Pure Over Aggression (POA); Instrumental Over Aggression (IOA); Pure Relational Aggression (PRA); Reactive Relational Aggression (RRA); Instrumental Relational Aggression (IRA); Emotional Perception (EP); Self-Emotional Management (SEM); Heteroemotional Management (HM); Emotional Utilisation (EU); Emotional Intelligence (EI); Violence (VI); Self-Concept (SC). Note: ** *p* ≤ 0.05; *** *p* ≤ 0.001.

## Data Availability

The data used to support the findings of current study are available from the corresponding author upon request.

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
