# Peer review of "Effectiveness of a Judo Intervention Programme on the Psychosocial Area in Secondary School Education Students"

_sports, 2023, doi:10.3390/sports11080140_

Round 1

Reviewer 1 Report

My recommendations are the following:

In abstract:

- lines 13-14 I recommend to mention only the relationships between them, not to present again the targeted dimensions.

- to describe descriptively what MC, SC, EI represent.

- It is not clear which evaluation tools were used, the Methods section.

- For the results, I recommend that you mention numerical values.

Line 116 I recommend deleting the word Design from the title of the subsection, this aspect does not emerge from the content.

Lines 132-135 mention that the questionnaire has 5 dimensions and describe only 4, I recommend the correction. I also recommend mentioning the value of Cronbach's alpha on each dimension.

Line 136 I recommend that you mention the value of Cronbach's alpha in both the pre-test and the post-test.

Line 142 I recommend that you write down the descriptions of what the TC represents.

Lines 142-144 when you mention sub-dimensions when sub-levels, I recommend clarification.

Line 147 which represents descriptive EL.

Line 157 mention the Cronbach's alpha value for the entire questionnaire for the two tests.

Line 164 I recommend that you mention when the study took place, the month and the year.

Lines 241-249 recommend that you do not duplicate the information with those presented in the tables, possibly to actually reproduce the difference between the arithmetic averages, not the test values.

Table 2, under which you mention what the calculated indices represent, the abbreviations.

I recommend that in table 2 you insert a new column in which to display the target size. It is difficult to identify what you are tracking just by mentioning the symbols of the subdimensions/sublevels, without a grouping by questionnaire or dimension. Ditto table 3 and 4.

I recommend that the Discussions section be reorganized on the targeted dimensions.

Author Response

REVIEWER 1

Comment 1

In abstract:

- lines 13-14: I recommend to mention only the relationships between them, not to present again the targeted dimensions.

- to describe descriptively what MC, SC, EI represent.

Response 1

Thank you very much for your comment. In this case the suggested changes have been applied.

Comment 2

- It is not clear which evaluation tools were used, the Methods section.

Response 2

Thank you very much for your comment. This section has been improved to make it easier to understand.

Comment 3

- For the results, I recommend that you mention numerical values.

Response 3

Thank you very much for your suggestion. As suggested by another reviewer, the authors have decided to leave the numerical value and standard deviation. This makes it easier for the reader to understand the article, since he/she can access the results more quickly.

Nevertheless, your suggestion will be applied to future research.

Comment 4

Line 116: I recommend deleting the word Design from the title of the subsection, this aspect does not emerge from the content.

Response 4

Thank you very much for your feedback. Your suggestion has been implemented.

Comment 5

Lines 132-135: mention that the questionnaire has 5 dimensions and describe only 4, I recommend the correction. I also recommend mentioning the value of Cronbach's alpha on each dimension.

Response 5

Thank you very much for your feedback. The missing item has been added

Comment 6

Line 136: I recommend that you mention the value of Cronbach's alpha in both the pre-test and the post-test.

Response 6

Thank you very much for your suggestion.

Cronbach's Alpha values for the pre-test and post-test have been added.

Comment 7

Line 14:  I recommend that you write down the descriptions of what the TC represents.

Response 7

Thank you very much for your suggestion.

The acronym has been removed.

Comment 8

Lines 142-144: When you mention sub-dimensions when sub-levels, I recommend clarification.

Response 8

Thank you very much for your suggestion.

A criterion has been applied to refer to the above.

Comment 9

Line 147: which represents descriptive EL.

Response 9

Thank you very much for your suggestion.

EL has been deleted and the word has been added.

Comment 10

Line 157: mention the Cronbach's alpha value for the entire questionnaire for the two tests.

Response 10

Thank you very much for your comment. The criteria you mentioned have been applied.

Comment 11

Line 164: I recommend that you mention when the study took place, the month and the year.

Response 11

Thank you very much for your comment. The following has been added to the research

Comment 12

Lines 241-249: Recommend that you do not duplicate the information with those presented in the tables, possibly to actually reproduce the difference between the arithmetic averages, not the test values.

Response 12

Thank you very much for your suggestion. After sharing your suggestion with the rest of the authors, we have decided to eliminate the values, leaving the table as the source of the query.

Comment 13

Table 2, under which you mention what the calculated indices represent, the abbreviations.

I recommend that in table 2 you insert a new column in which to display the target size. It is difficult to identify what you are tracking just by mentioning the symbols of the subdimensions/sublevels, without a grouping by questionnaire or dimension. Ditto table 3 and 4.

Response 13

Thank you very much for your comment.

The authors have proceeded to facilitate the understanding of table 2.

However, for tables 3 and 4, the results are grouped according to the variables. This has been done by default by the program. We have also tried to eliminate the abbreviations, but the table was very poorly presented. We have therefore opted to keep the format and add the abbreviations at the top of tables 3 and 4.

Comment 14

I recommend that the Discussions section be reorganized on the targeted dimensions.

Response 14

Thank you very much for your suggestion. The discussion has been reorganized.

Reviewer 2 Report

Dear Authors

You have written an interesting study focusing on the investigation of judo intervention program on the motivational climate towards sport, aggressive behavior, emotional intelligence and self-concept in secondary school students.

However, the overall methodology is extremely poor. For start, the specific judo knowledge level physical education teachers. So how long was the educational program for them? Did all PE teachers reach the adequate level of knowledge, who assessed them - how many examiners, and what was their agreement? The description of training protocol - intervention is very poor and in this state, it can not be reproducible. Which elements were taught - for which kyu degree, etc?

Physical activity levels of participants that could affect the outcome are not reported. This means that the inclusion criteria were poorly managed. Also, the control group is missing for the study to have any relevant conclusion.

The discussion is very poor and does not connect well to the available literature.

Therefore, I have to reject this paper.

Moderate editing of the English language required

Author Response

REVIEWER 2

Comment 1

Dear Authors

You have written an interesting study focusing on the investigation of judo intervention program on the motivational climate towards sport, aggressive behavior, emotional intelligence and self-concept in secondary school students.

However, the overall methodology is extremely poor. For start, the specific judo knowledge level physical education teachers. So how long was the educational program for them? Did all PE teachers reach the adequate level of knowledge, who assessed them - how many examiners, and what was their agreement? The description of training protocol - intervention is very poor and in this state, it can not be reproducible. Which elements were taught - for which kyu degree, etc?

Physical activity levels of participants that could affect the outcome are not reported. This means that the inclusion criteria were poorly managed. Also, the control group is missing for the study to have any relevant conclusion.

The discussion is very poor and does not connect well to the available literature.

Therefore, I have to reject this paper.

Response 1

Thank you very much for your comments.

The authors honestly disagree with many of the comments you convey to us. Initially, when you cite that the methodology of the study is poor, we do not understand why. In this case a quasi-experimental study in a single intervention group has been approached, there being recent research similar to this in the journal Sports:

Rejeki, P.S.; Pranoto, A.; Rahmanto, I.; Izzatunnisa, N.; Yosika, G.F.; Hernaningsih, Y.; Wungu, C.D.K.; Halim, S. The Positive Effect of Four-Week Combined Aerobic-Resistance Training on Body Composition and Adipokine Levels in Obese Females. Sports 2023, 11, 90. https://doi.org/10.3390/sports11040090

Regarding the training of the physical education teachers, more data have been added to the study, however they are attached again: The training received by the physical education teachers focused on the technique and dynamics of Judo classes. The training period for the teachers lasted three months. Regarding the technical training, the teachers were trained in the analysis and performance of different keys (O-Soto-Gari, Yoko-Shiho-Gatame Onko-Shiho-Gatame and Uki-Goshi). Regarding class dynamics, the teachers were trained in Judo didactics, receiving classes from a second Dan belt, specialized in the didactics of this sport. Likewise, the intervention program was based on the curriculum that governs compulsory secondary education in Spain, so that this intervention program was contextualized within the Spanish educational law.

Likewise, for this intervention to be justified within the school curriculum of the secondary education stage, the level had to be initiation, being this carried out through games and the teaching of some techniques. Non-compliance with the curriculum would imply an illegality in the teaching of the content.

Similarly, the inclusion criteria were defined by the educational environment, since all the young people in the school year participated. If any student is discriminated against, it is because he/she has not been authorized to participate by his/her legal guardians. Likewise, the above statement together with the presence of only one intervention group does not jeopardize either the integrity or the honesty of the research presented.

Finally, the data related to physical activity remain in the background since what is of interest is to evaluate whether the intervention program has been effective in increasing the psychosocial levels to be studied and decreasing the levels of anxiety.

Reviewer 3 Report

Dear Authors,

I would like to express my gratitude regarding the opportunity to review this manuscript.

At this stage the document requires improvements, below with line indication:

6-10 – Please include author´s initials close to emails.

39 – Please change “&” by “and”. Same in lines 52, 54. Please carefully revise all manuscript.

64 – Citation number is not according to references. Please carefully revise all references and citations format and numbers.

119 – Please standardize the decimals of M ± SD.

123-124 – Please insert line spacing. Please revise this detail throughout the manuscript.

123 – Please describe exclusion criteria, written informed consents, and other details related to the sample (weekly routines, active/sedentary? and others).

179 – Please describe with more detail the program (duration, time of day, routines, environmental conditions, equipment, who conducted the sessions and evaluations (academic background, experience). All details should be described.

189 – Please indicate “IBM SPSS” city and country.

189 – Please address the sample size of the study.

212-231 – Behind the figure numbers are observed, please format.

238 – Please format the table considering the journal template and instructions for authors. Please also standardize the decimals in the %´s.

250 – Please format the table considering the journal template and instructions for authors. “[.17; .3]” is not clear.

258 – Please confirm if the “p” should be in italic throughout the manuscript.

272 – The table 3 footnote is missing. Please insert.

284 - The table 4 footnote is missing. Please insert.

286 – The discussion should present in the beginning (first paragraph) the aim of the study and the main findings, to afterward analyse these comparing with the literature, in the following paragraphs.

320 – Please correct the spaces.

285-330 – The discussion section is too short, please consider improving the content and quality.

332 – Please consider more direct messages in the conclusions section, if possible, with practical applications.

339-351 – Please revise, considering the journal template.

352 - Please carefully revise all the references format. They are not according to the journal template.

A careful final reading after completing V2 considering English improvement is suggested.

Moderate editing of English language required.

Author Response

REVIEWER 3

Dear Authors,

I would like to express my gratitude regarding the opportunity to review this manuscript.

At this stage the document requires improvements, below with line indication:

Comment 1

6-10 – Please include author´s initials close to emails.

Response 1

Thank you very much for your comment.

The change you suggested has been implemented.

Comment 2

39 – Please change “&” by “and”. Same in lines 52, 54. Please carefully revise all manuscript.

Response 2

Thank you very much for your comment.

The change you suggested has been implemented.

Comment 3

64 – Citation number is not according to references. Please carefully revise all references and citations format and numbers.

Response 3

Thank you very much for your comment.

The citations and references have been revised and the errors have been corrected.

 Comment 4

119 – Please standardize the decimals of M ± SD.

Response 4

Thank you very much for your comment.

Your comment has been implemented.

Comment 5

123-124 – Please insert line spacing. Please revise this detail throughout the manuscript.

Response 5

Thank you very much for your comment.

The entire document has been revised to avoid gaps.

Comment 6

123 – Please describe exclusion criteria, written informed consents, and other details related to the sample (weekly routines, active/sedentary? and others).

Response 6

Thank you very much for your comment. More details related to inclusion and exclusion criteria have been added.

 Comment 7

179 – Please describe with more detail the program (duration, time of day, routines, environmental conditions, equipment, who conducted the sessions and evaluations (academic background, experience). All details should be described.

Response 7

Thank you very much for your comment. More details related to the teacher training process and the intervention program itself have been added.

 Comment 8

189 – Please indicate “IBM SPSS” city and country.

 Response 8

Thank you very much for your comment. The requested information has been added

Comment 8

189 – Please address the sample size of the study.

 Response 8

Thank you very much for your suggestion. The sample size has not been added because it has not been possible to access the total number of students enrolled in the province where the intervention program was carried out

Comment 9

212-231 – Behind the figure numbers are observed, please format.

Response 9

Thank you very much for your comment. Numbers have been removed.

Comment 10

238 – Please format the table considering the journal template and instructions for authors. Please also standardize the decimals in the %´s.

Response 10

Thank you very much for your comment. Your comments have been applied.

 Comment 11

250 – Please format the table considering the journal template and instructions for authors. “[.17; .3]” is not clear.

Response 11

Thank you very much for your comment. Your comments have been applied.

Comment 12

258 – Please confirm if the “p” should be in italic throughout the manuscript.

Response 12

Thank you very much for your comment. P should be in Italics.

Comment 13

272 – The table 3 footnote is missing. Please insert.

Response 13

Thank you very much for your comment. Table 3 footnote has been included.

Comment 14

284 - The table 4 footnote is missing. Please insert.

Response 14

Thank you very much for your comment. Table 4 footnote has been included.

Comment 15 

286 – The discussion should present in the beginning (first paragraph) the aim of the study and the main findings, to afterward analyse these comparing with the literature, in the following paragraphs.

Comment 16

320 – Please correct the spaces.

Response 16

Thank you very much for your comment. Spaces have been corrected

Comment 17

285-330 – The discussion section is too short, please consider improving the content and quality.

Comment 18

332 – Please consider more direct messages in the conclusions section, if possible, with practical applications.

Response 18

Thank you very much for your comment. The wording of the conclusions has been reworded. Likewise, the practical applications have been included together with the limitations and future perspectives. 

Comment 19

339-351 – Please revise, considering the journal template.

Response 19

Thank you very much for your comment. The criteria set out in the template have been complied with at all times.

Comment 20

352 - Please carefully revise all the references format. They are not according to the journal template.

Response 20

Thank you very much for your comment. The correct reference format has been applied.

Reviewer 4 Report

the paper in its abstract should present the context of the study

the paper in its abstract should present the context of the study

In the same way, in the participants section, the authors should indicate the province and the city of the autonomous community of Andalusia in Spain.

Since the authors deal with a topic of current interest and interest such as the psychosocial area and within this area of emotional intelligence, I beg you to see and consider these articles since they deal with emotional development in physical education

Zamorano-García, M., Gil-Madrona, P., Prieto-Ayuso, A., & Zamorano García, D. (2018). Emociones generadas por distintos tipos de juegos en clase de educación física / Generated Emotions by Various Types of Games in Physical Education. Revista Internacional de Medicina y Ciencias de la Actividad Física y el Deporte, 18(69), 1-26 Http://cdeporte.rediris.es/revista/revista69/artemoc iones869.htm doi: https://doi.org/10.15366/rimcafd2018.69.001

Gil-Madrona, P., Samalot-Rivera, A., & Kozub, F. M. (2016). Acquisition and transfer of values and social skills through a physical education program focused in the affective domain. Motricidade, 12(3), 32–38. Retrieved from https://doi.org/10.6063/motricidade.6502

Gil-Madrona, P., Pascual-Francés, L., Jordá-Espi, A., Mujica-Johnson, F., & Fernández-Revelles, A. B. (2020). Affectivity and Motor Interaction in Popular Motor Games at School. Apunts. Educación Física y Deportes, 139, 42-48. https://doi.org/10.5672/apunts.2014-0983.es.(2020/1).139.06

The subject of study is of great interest, the work is good, the methodological procedure is good and the presentation of results is also good.

Author Response

REVIEWER 4

Comment 1

The paper in its abstract should present the context of the study

Response 1

Thank you very much for your comment. The suggested changes have been implemented.

Comment 2

In the same way, in the participants section, the authors should indicate the province and the city of the autonomous community of Andalusia in Spain.

Response 2

Thank you very much for your comment. The suggested information has been added.

Comment 3

Since the authors deal with a topic of current interest and interest such as the psychosocial area and within this area of emotional intelligence, I beg you to see and consider these articles since they deal with emotional development in physical education

Zamorano-García, M., Gil-Madrona, P., Prieto-Ayuso, A., & Zamorano García, D. (2018). Emociones generadas por distintos tipos de juegos en clase de educación física / Generated Emotions by Various Types of Games in Physical Education. Revista Internacional de Medicina y Ciencias de la Actividad Física y el Deporte, 18(69), 1-26 Http://cdeporte.rediris.es/revista/revista69/artemoc iones869.htm doi: https://doi.org/10.15366/rimcafd2018.69.001

Gil-Madrona, P., Samalot-Rivera, A., & Kozub, F. M. (2016). Acquisition and transfer of values and social skills through a physical education program focused in the affective domain. Motricidade, 12(3), 32–38. Retrieved from https://doi.org/10.6063/motricidade.6502

Gil-Madrona, P., Pascual-Francés, L., Jordá-Espi, A., Mujica-Johnson, F., & Fernández-Revelles, A. B. (2020). Affectivity and Motor Interaction in Popular Motor Games at School. Apunts. Educación Física y Deportes, 139, 42-48. https://doi.org/10.5672/apunts.2014-0983.es.(2020/1).139.06

The subject of study is of great interest, the work is good, the methodological procedure is good and the presentation of results is also good.

Response 3

Thank you very much for your suggestion. The requested bibliography has been added.

Round 2

Reviewer 1 Report

No comments

Author Response

Thank you very much for your assessment

Reviewer 2 Report

Dear Authors

Thank you for addressing the comments and feedback and resubmitting your paper. The quality and clarity improved. I am satisfied with your responses.

However, there are still some parts where additional info is needed that affect the reproducibility of your study. 

Line 194 - can you explain which judo element is Onko-Shiho-Gatame or is this a typo? Also, what about the breakfall techniques - were the PE teachers also educated on this as in the training description you mention this? Elaborate and report. Please provide more details regarding the 3 months of education of PE teachers, how many hours in total this education lasted, how many hours of lectures, and how many hours of practicals on mats. Who evaluated their knowledge? Please be specific.

Lines 203-213 / Again - how can someone replicate your study with this description? Nobody.

So the study lasted 3 months, 2 trainings per week, 50 min per training - So in total 24 trainings.

And what about everything else? There is no detailed info regarding your intervention program and I don't understand how can you submit a paper without it!

Every training session has a warm-up, main part and cool-down. Please specify the durations of these parts used in your programs. Also, state the exact judo elements which were used in the teaching process. From your description, I can only assume the same ones PE teachers were educated on, but this is not sure as you mention other elements in the description. So what was it then? Be specific. You can add this as a supplement.

Like in PE, you have the main outcome in mind, like overhead pass, salto, etc. What were your aims in this intervention: to teach breakfalls, standing techniques, transition to Osaekomi-waza, able to have standing fight-match.

Discussion:

The first paragraph mentioned that judo is dangerous. However, none of the referenced papers, never used the word ''dangerous'' once in their paper or in connection to judo. So, rephrase this part and do not report things out of context! Also, this does not align with your aims at the end of the introduction. Please amend your text.

Lines 347-349 / report percentages of these MA participations.

Line 350 - so did you do that / teach breakfalls? Poor connection of your program with the literature.

Also, please clearly state in the introduction that judo is a combat sport & martial art as there is a difference.

Limitations section - what about PE teachers not being judokas and perhaps their level of knowledge impacted the performance of the program? Please add this as a limitation.

Practical application - can you be more specific/primary education - add breakfalls and try to be more specific - how many hours per year in PE classes would be beneficial or recommended by you, to be focused on judo elements.

Your paper has potential, however, it needs better clarity on the program so it can be replicated. This will add weight to your study. I hope you see this as constructive feedback.

Kind regards

Minor editing of the English language required

Author Response

REVIEWER 2

Thank you for addressing the comments and feedback and resubmitting your paper. The quality and clarity improved. I am satisfied with your responses.

However, there are still some parts where additional info is needed that affect the reproducibility of your study.

Comment 1

Line 194 - can you explain which judo element is Onko-Shiho-Gatame or is this a typo? Also, what about the breakfall techniques - were the PE teachers also educated on this as in the training description you mention this? Elaborate and report. Please provide more details regarding the 3 months of education of PE teachers, how many hours in total this education lasted, how many hours of lectures, and how many hours of practicals on mats. Who evaluated their knowledge? Please be specific.

Response 1

Thank you very much for your comments. Regarding the first Onko-Shiho-Gatame, it was a translation error. It refers to Tate-Shiho-Gatame. A greater amount of detail has been added. Also in the previous revision it was already specified who assessed teachers' learning.

Comment 2

Lines 203-213 / Again - how can someone replicate your study with this description? Nobody.

So the study lasted 3 months, 2 trainings per week, 50 min per training - So in total 24 trainings.

And what about everything else? There is no detailed info regarding your intervention program and I don't understand how can you submit a paper without it!

Every training session has a warm-up, main part and cool-down. Please specify the durations of these parts used in your programs. Also, state the exact judo elements which were used in the teaching process. From your description, I can only assume the same ones PE teachers were educated on, but this is not sure as you mention other elements in the description. So what was it then? Be specific. You can add this as a supplement.

Response 2

Thank you for your comment. More details have been added on the distribution of the sessions and how time was managed in the sessions.

Comment 3

Like in PE, you have the main outcome in mind, like overhead pass, salto, etc. What were your aims in this intervention: to teach breakfalls, standing techniques, transition to Osaekomi-waza, able to have standing fight-match.

Response 3

Thank you very much for your suggestion. The objectives of the intervention programme have been added

Comment 4

Discussion:

The first paragraph mentioned that judo is dangerous. However, none of the referenced papers, never used the word ''dangerous'' once in their paper or in connection to judo. So, rephrase this part and do not report things out of context! Also, this does not align with your aims at the end of the introduction. Please amend your text.

Response 4

Thank you for your comment. The suggested changes have been implemented

Comment 5

Lines 347-349 / report percentages of these MA participations.

Response 5

Thank you for your comment. The suggested changes have been implemented

Comment 6

Line 350 - so did you do that / teach breakfalls? Poor connection of your program with the literature.

Response 6

Thank you for your comment. No breakfalls have been taught

Comment 7

Also, please clearly state in the introduction that judo is a combat sport & martial art as there is a difference.

Response 7

Thank you for your comment. Your suggestion has been added

Comment 8

Limitations section - what about PE teachers not being judokas and perhaps their level of knowledge impacted the performance of the program? Please add this as a limitation.

Response 8

Thank you for your comment. Your suggestion has been added

Comment 9

Practical application - can you be more specific/primary education - add breakfalls and try to be more specific - how many hours per year in PE classes would be beneficial or recommended by you, to be focused on judo elements.

Response 9

Thank you for your comment. Your suggestion has been added

Comment 10

Your paper has potential, however, it needs better clarity on the program so it can be replicated. This will add weight to your study. I hope you see this as constructive feedback.

Response 10

 Thank you for your comment. We hope we have improved the research

Reviewer 3 Report

Dear Authors,

Thank you for considering my suggestions and incorporating them into the manuscript, which is globally improved, congratulations.

Below are suggestions related to this last version (v2), with line indication.

123 – SD suggested with 2 decimals. 288 – Table 2, criteria 2 decimals in SD.

182 – “physical education” is presented in the manuscript around 10x, please consider abbreviating “PE”. Same suggestion regarding “physical activity” – “PA”.

203-213 – The inclusion and exclusion criteria were developed, thank you for considering my suggestion. Nevertheless, weekly routines, (active/sedentary/ number of training sessions) and other information related to the sample characterization were not included (only table 1 presents some information). Please consider including all details related to the subjects’ characterization, intervention program, and data collection.

221 – Comment 8 – Sample size description was not included in the manuscript. The suggestion was not related to the number of students enrolled but to the data calculated in GPower software.

239 – Please consider improving the figure quality.

310 – The footnote is above the line. Please correct.

318 – “Concept”, please revise if not in lowercase.

333 – The direction column presents values with space and others not. Please standardize.

342-367 – Please consider standardizing the paragraph size. Please consider this suggestion not only in these lines but also throughout the manuscript.

370, 375 – Two phrases start with “Likewise”. Please consider improving the English quality throughout the manuscript. Line 391 again “Likewise”.

402 – “Judo” in uppercase but sometimes in the manuscript in lowercase. Please standardize.

420, 421 - Two phrases start with “In addition”. Please consider improving the English quality throughout the manuscript.

425 – Please remove “. Same in 431, 435, and 437. From 425 to 437 please remove the “and insert end points.

438 – Please double-check all the references’ format. Some examples: L 465 – pages missing; 469 – Year not in bold; 471 – italic missing.

Moderate editing of English language required.

Author Response

REVIEWER 3

Comment 1

123 – SD suggested with 2 decimals. 288 – Table 2, criteria 2 decimals in SD.

Response 1

Thank you for your comment. Criteria have been unified

Comment 2

182 – “physical education” is presented in the manuscript around 10x, please consider abbreviating “PE”. Same suggestion regarding “physical activity” – “PA”.

Response 2

Thank you for your comment. Synonyms have been sought to avoid repetition of the words quoted.

Comment 3

203-213 – The inclusion and exclusion criteria were developed, thank you for considering my suggestion. Nevertheless, weekly routines, (active/sedentary/ number of training sessions) and other information related to the sample characterization were not included (only table 1 presents some information). Please consider including all details related to the subjects’ characterization, intervention program, and data collection.

Response 3

Thank you very much for your comment. Only these results can be included in table 1. The variables presented in the table were not measured after the intervention programme. In this case the number of routines per week is presented in the procedure section.

Comment 4

221 – Comment 8 – Sample size description was not included in the manuscript. The suggestion was not related to the number of students enrolled but to the data calculated in GPower software.

Response 4

Thank you very much for your comment. Cohen's d has been used in this case. These data are available in table 2

Comment 5

239 – Please consider improving the figure quality.

Response 5

Thank you for your suggestion. The image quality has been improved

Comment 6

310 – The footnote is above the line. Please correct.

Response 6

Thank you for your suggestion. It has been corrected

Comment 7

318 – “Concept”, please revise if not in lowercase.

Response 7

Thank you for your comment. That word is capitalised.

Comment 8

333 – The direction column presents values with space and others not. Please standardize.

Response 8

Thank you for your suggestion. It has been corrected

Comment 9

342-367 – Please consider standardizing the paragraph size. Please consider this suggestion not only in these lines but also throughout the manuscript.

Response 9

Thank you for your comment. The size of the paragraphs has been shortened.

Comment 10

370, 375 – Two phrases start with “Likewise”. Please consider improving the English quality throughout the manuscript. Line 391 again “Likewise”.

Response 10

Thank you for your comment. Synonyms for these words have been searched for.

 Comment 11

402 – “Judo” in uppercase but sometimes in the manuscript in lowercase. Please standardize.

Response 11

Thank you for your comment. Criteria have been unified

 Comment 12

420, 421 - Two phrases start with “In addition”. Please consider improving the English quality throughout the manuscript.

Response 12

 Thank you for your comment. Criteria have been unified

Comment 13

425 – Please remove “. Same in 431, 435, and 437. From 425 to 437 please remove the “and insert end points.

Response 13

Thank you for your comment. The suggested changes have been implemented

Comment 14

438 – Please double-check all the references’ format. Some examples: L 465 – pages missing; 469 – Year not in bold; 471 – italic missing.

Response 14

Thank you for your comment. If you consult the referencing rules, the years are in bold. We invite you to check. There are some references that do not have pages, but an identifying number. Italics have been added